# Hormonal Profile in Response to an Empathic Induction Task in Perpetrators of Intimate Partner Violence: Oxytocin/Testosterone Ratio and Social Cognition

**DOI:** 10.3390/ijerph19137897

**Published:** 2022-06-27

**Authors:** Javier Comes-Fayos, Ángel Romero-Martínez, Isabel Rodríguez Moreno, María Carmen Blanco-Gandía, Marta Rodríguez-Arias, Marisol Lila, Concepción Blasco-Ros, Sara Bressanutti, Luis Moya-Albiol

**Affiliations:** 1Department of Psychobiology, University of Valencia, 46010 Valencia, Spain; javier.comes@uv.es (J.C.-F.); angel.romero@uv.es (Á.R.-M.); isabel.rodriguez-moreno@uv.es (I.R.M.); marta.rodriguez@uv.es (M.R.-A.); concepcion.blasco@uv.es (C.B.-R.); sarabressanutti@gmail.com (S.B.); 2Department of Psychology and Sociology, University of Zaragoza, 50009 Zaragoza, Spain; mcblancogandia@unizar.es; 3Department of Social Psychology, University of Valencia, 46010 Valencia, Spain; marisol.lila@uv.es

**Keywords:** emotion induction, empathy, oxytocin, perspective taking, oxytocin–testosterone ratio, testosterone–cortisol ratio

## Abstract

Empathy deficits have been proposed to be an important factor for intimate partner violence (IPV). IPV perpetrators have shown a differential change in salivary oxytocin (sOXT), testosterone (sT), and cortisol (sC), following empathic and stress tasks, compared to non-violent men. However, the influence of empathic deficits in those hormones after an emotion-induction task in IPV perpetrators remains unclear. We analyzed the effects of an empathic induction task on endogenous sOXT, sT and sC levels, as well as their hormonal ratios, in IPV perpetrators (*n* = 12), and compared them to controls (*n* = 12). Additionally, we explored the predictive capacity of empathy-related functions (measured with the interpersonal reactivity index) in the hormonal responses to the task. IPV perpetrators presented lower sOXT changes and higher total sT levels than controls after the task, lower sOXT/T change and total sOXT/T levels, as well as higher total sT/C levels. Notably, for all participants, the lower the perspective taking score, the lower the total sOXT levels and sOXT changes and the higher the sT changes were. Low perspective taking also predicted smaller sOXT/T and sOXT/C changes in the empathic induction task, and higher total sT/C levels for all participants. Therefore, our results could contribute to furthering our ability to focus on new therapeutic targets, increasing the effectiveness of intervention programs and helping to reduce IPV recidivism in the medium term.

## 1. Introduction

According to the World Health Organization [1], intimate partner violence (IPV) is a major public health challenge worldwide, encompassing a wide variety of violent behaviors and resulting in significant economic, social, and human costs. To prevent IPV, a growing number of studies have focused on the profile of IPV perpetrators, identifying various characteristics, such as anger management difficulties, impulsivity, sexism, substance abuse, childhood trauma, or lack of community support, as particularly relevant with regards to the severity and recidivism of IPV [2,3,4]. However, the need to approach IPV from a multidimensional biopsychosocial framework has been highlighted, bringing together the psychosocial factors of IPV with biological markers that provide data that are less susceptible to bias and manipulation than self-reported questionnaires [5,6,7,8,9].

Difficulties in empathy, defined as the ability to understand and share another person’s emotional experience, seemed to play a key role in the maladaptive behavioral regulation of IPV perpetrators [10,11]. Indeed, impaired empathy has been linked to the occurrence of violent behaviors [12,13], including IPV perpetration [14,15]. Specifically, difficulties in understanding the thoughts and feelings of the partner could predispose someone to IPV by promoting affective distress when dealing with emotional conflicts [16,17,18,19]. Furthermore, the comparison of IPV perpetrators with controls revealed that the former tends to present lower scores in perspective taking (ability to adopt the psychological point of view of others) and empathic concern (emotional response of compassion to the suffering of others) and show higher personal distress (discomfort in reacting to the emotions of others) [20,21].

Nevertheless, the importance of using natural approaches in conjunction with self-report questionnaires has been emphasized when dealing with a complex construct, such as empathy [22,23]. Yet, in many cases, it is difficult to control empathy in its natural setting without altering the variables of the target situation. In this regard, emotion induction tasks help to evoke the required affective experience in an experimental situation through various methods (e.g., images, sounds, or both, among others) [24,25]. In fact, these methods can be centered on one’s own or another’s affective state, enabling in situ characterization of empathy and gaining access to the natural study of other variables of interest, such as associated hormone levels [26].

Changes in oxytocin (OXT), testosterone (T) and cortisol (C) endogenous levels could affect functions important for empathy, including socioaffective information processing, social affiliation, or stress management [27,28,29,30]. Accordingly, increases in endogenous OXT levels, both in plasma (pOXT) and saliva (sOXT), have been observed in a normative population following an empathic induction task performed through watching an emotion-provoking video [24,25]. In particular, Barraza and Zak [24] found that self-reported affective empathy predicted an increase in pOXT levels following the empathic induction task, whereas personal distress predicted a decrease in pOXT levels. Meanwhile, Procyshyn et al. [25] found that increased sOXT, following the same empathic induction task, was related to improved perspective taking. In IPV perpetrators, a study concluded that IPV perpetrators presented lower sOXT levels than controls at a specific moment after an empathic induction task [26]. Thus, these results posit a positive relationship between OXT and socioaffective functions that facilitate empathy and suggest that the relationship between OXT and IPV could be mediated by difficulties in these functions.

Regarding T, Chen et al. [31] found that baseline plasmatic T (pT) levels were negatively correlated with empathic concern and perspective-taking in a non-forensic population. Moreover, decreases in salivary T (sT) levels were observed in college students following an empathic induction task [25]. Baseline sT levels have also been negatively related to the ability to infer the thoughts and feelings of others in a role-playing paradigm [32]. Consistently, Romero-Martínez et al. [20,33] obtained that high baseline sT levels were associated with worse facial emotion recognition in IPV perpetrators. All of this, taken together, suggests that T could be an important modulator of violent behavior through its buffering effects on the processing of socioaffective information.

In relation to C, it seems that C maintains a negative association with empathic-related variables, concretely, with emotion decoding abilities. Accordingly, endogenous C rises after hydrocortisone administration reduces the ability to correctly decode emotional facial expressions [21]. Regarding IPV, there is a gap in the scientific literature for assessing whether emotion induction tasks modified the C response in IPV perpetrators. In fact, only two studies have assessed their response to acute laboratory stressors [20,30]. In this sense, they concluded that IPV perpetrators showed a lower sC response than controls after being submitted to an acute laboratory stressor, failing to find significant associations between empathic variables and sC levels in them [20].

To fully understand a phenomenon as complex as empathy or violence proneness, the interrelationships between the above-mentioned hormones should be incorporated. Accordingly, studies have shown facilitative effects of OXT and damping effects of T on the same cognitive and behavioral phenotypes. Specifically, comparable diametrical effects have been observed in empathy-relevant tasks, such as emotional recognition, theory of mind or self-reported empathy [27]. On the other hand, although C is the main stress hormone, OXT also responds under stressful situations [34]. Evidence has been provided that intranasal administration of OXT decreases sC during an interpersonal challenge [35,36], resulting in lower social stress, which has been linked to enhanced empathic function. Nevertheless, a meta-analysis concluded that the relationship between the endogenous levels of both hormones tends to be positive but small [37]. Conversely, the administration of OXT, T and C on facial emotion recognition has been analyzed, concluding that emotional accuracy seemed to improve after increasing OXT, but decreased when T and/or C increased [21].

Regarding T and C interaction, a higher sT/C ratio was predictive of lower empathy scores in healthy young adults [38]. Furthermore, higher sT/C ratios have been associated with aggression rates within heterosexual couples [39]. In fact, it seems that IPV perpetrators tend to present higher sT/C ratios than controls in response to acute laboratory stressors [20,30]. These results are consistent with the dual hormone hypothesis, which suggests that T is positively associated with aggression and negatively with empathy, particularly in individuals with low C levels [40]. Unfortunately, there is a gap in the scientific literature for assessing the interaction between the three above-mentioned hormones in IPV perpetrators, specifically in response to emotion induction tasks.

Consequently, the present study pursues three major objectives. We first aim to compare self-reported empathy in IPV perpetrators and controls. Based on that stated in this field of research [10,11], we expect to find worse perspective taking and lower empathic concern, as well as higher personal distress, in IPV perpetrators compared to the control group [20].

Second, we intend to analyze differences in endogenous sOXT, sT, and sC levels, as well as the quotient between levels of the above-mentioned hormones (hormonal ratios), of a group of IPV perpetrators compared to controls following an empathic induction task. As stated before in this field of research [25,26,27,36,41], we hypothesize that IPV perpetrators will display lower levels of sOXT and sC and higher levels of sT after the empathic induction task compared to the control group. Furthermore, IPV perpetrators will also show lower sOXT/T and sOXT/C ratios and higher sT/C ratios in response to this task than controls.

Finally, we also aim to explore whether the empathic abilities of each group explain their hormone levels following the empathic induction task. Due to the relationship between these hormones and empathy [21,27,38,42], we hypothesize that perspective-taking and empathic concern scores would especially predict the IPV perpetrators’ hormonal levels in response to this task. Accordingly, the lower the empathic abilities of IPV perpetrators, the lower the sOXT response and the higher the sT/C ratio.

## 2. Materials and Methods

### 2.1. Participants

From an initial total sample of 25 participants, one IPV perpetrator was excluded from the study because his hormonal levels exceeded the standard deviation of the total sample by 2.5 or more. Hence, the total sample consisted of 24 healthy male volunteers (12 IPV perpetrators and 12 controls).

IPV perpetrators were recruited among the participants of the CONTEXTO Program, a community-based psychoeducational intervention program (mandatory for men convicted of gender-based violence, with sentences of up to two years imprisonment), conducted at the Faculty of Psychology of the University of Valencia, before starting the intervention program. To be a part of this study, it was necessary that IPV perpetrators receive a suspended sentence for gender-based violence under the condition of attending the CONTEXTO intervention program. In addition, participants had to not have a diagnosis associated with a psychiatric, neurological or substance use disorder, and had to understand and speak fluent Spanish [3].

Participants of the control group were recruited in Valencia (Spain), establishing contact through advertisements on social networks. After conducting a telephone interview, the participants who met the requirements were selected (e.g., absence of mental disorders, and endocrine alterations). They were volunteers with similar sociodemographic (age, marital status, educational level and annual income) characteristics and body mass index (BMI), as well as not having criminal records.

All participants were informed of the study and, voluntarily, gave their written informed consent. Upon completion of the study, all participants received a financial compensation of EUR 40.

The experiment was performed in accordance with the Declaration of Helsinki and was approved by the Ethics Committee of the University of Valencia (procedure number: H1538385543901).

### 2.2. Procedure

The procedure was conducted in the laboratories of the Department of Psychobiology of the University of Valencia in a single session of approximately 2 h. The laboratory protocol was performed between 16:00 and 19:00 h, as sC levels are more stable within this period of time [20]. Additionally, the session was held in a room that was kept at a constant temperature (21 °C) and humidity. First, the informed consent was signed, sociodemographic and BMI variables were measured, and the empathy questionnaire was administered.

Afterwards, baseline hormonal levels were collected. The hormonal samples were always obtained following the same order: sOXT and sC first, followed by sT. Then, the empathic induction task instructions were provided to the participants and, immediately, the anticipatory sOXT, sC, and sT were collected. Immediately after completing the empathic induction task, the post-empathic sOXT, sC, and sT were taken. Three additional hormone samples were collected at +20 min, +40 min and +60 min. Finally, the participants were accompanied and dismissed.

### 2.3. Measures

#### 2.3.1. Empathic Induction Task

Following the guidelines of previous studies [24,25,26], the empathic induction task was carried out using a battery of emotion-provoking videos validated for the Spanish population (PIE) [43]. Because the procedure is focused on negative emotional content, four scenes where the main character was receiving violence were selected among the entire battery based on their high arousal and negative affect. The selection of these scenes was made considering the suggestions from Gracia et al. [44] and Romero-Martínez et al. [45], who reported the existence of cognitive and affective alterations in IPV perpetrators when facing IPV audio–visual content. Hence, to avoid a gender bias, we combined two scenes where female characters received violence and two scenes where violence was received by male characters. Each participant underwent the empathic induction task by viewing the scenes (with a mean of 1′ 21″) on a 75″ screen. Prior to the visualization of the scenes, participants were instructed to actively put themselves in the place of each main character (previously indicated). Following guidelines from Schaefer et al. [46], participants were asked to relax for one minute before watching each scene.

#### 2.3.2. Empathy Questionnaire

For the measurement of empathy, the Spanish adaptation [47] of the interpersonal reactivity index (IRI) [48] was used, which evaluates perspective taking (the tendency to spontaneously adopt the point of view of others), fantasy (the subject’s imaginative capacity to put themselves in fictitious situations), empathic concern (feelings of compassion, worry and affection in the face of the discomfort of others) and personal distress (feelings of anxiety and discomfort that the subject manifests when observing the negative experiences of others). The items were rated from 1 (does not describe me well) to 5 (describes me well). For this study, Cronbach’s alpha was 0.77.

#### 2.3.3. Hormonal Levels

Saliva was directly collected from the mouth using Salivettes for OXT and C (Sarstedt, Rommelsdorf, Germany) and sterile glass tubes for T measurements. In all cases, participants were informed about the need to follow the instructions for saliva sampling to obtain meaningful data, and samples were collected in the same order, OXT-C then T, and frozen at −20 °C until analysis.

Salivary T levels were assessed by chemiluminescence immunoassays using testosterone saliva ELISA kits (Diagnostics Biochem Canada Inc., London, ON, Canada). Intra- and interassay coefficients of variation were 3.98% and 7.98%, respectively, indicating good reproducibility. On the other hand, salivary C levels were determined by radioimmunoassay using Coat-to-Count cortisol kits (DPC-Siemens Medical Solutions Diagnostics, Chennai, India) with 1.4 nmol/L sensitivity.

Salivary OXT were measured using the commercial Oxytocin EIA kit (Arbor Assays, Inc., Ann Arbor, MI, USA; ref: K048). We followed the procedure described in previous studies [49,50,51,52]. The samples were centrifuged 5 min × 1000 *g* and then, from each saliva sample, 1 mL of supernatant was stored and frozen at −40 °C. Then, the samples were lyophilized (Modulyo Freeze Dryers, Thermo Electron Corporation, Waltham, MA, USA) for approximately 15 h, until they were dehydrated [53]. These samples were reconstituted in 250 μL of assay buffer, yielding a concentration four times greater than the original, allowing them to fall within the sensitivity range of the kit and be detectable on the standard curve. Remarkably, it was appreciated that sOXT correlates better than pOXT with concentrations in the cerebrospinal fluid [54]. Neuropeptide cross-reactivity was reported by Arbor assays as <0.001% and the detection limit was 11 pg/mL.

All samples were analyzed in duplicate, and those from the same subject were included in the same assay. Although the criterion for measurement replication was a coefficient of variation between duplicates of 8%, the maximum intra- and inter-assay coefficients of variation obtained were 4.3% and 5.2%, respectively.

### 2.4. Data Analysis

After checking the data for normality with the Shapiro–Wilk test (*p* < 0.05), the non-normal data were transformed based on a Napierian logarithm (logn). Afterwards, *t*-tests were performed for assessing group (IPV perpetrators and controls) differences for anthropometric variables, empathy scores and baseline hormonal levels. Effect sizes for the between-group differences were calculated using Cohen’s d [55]. A chi-squared analysis was also performed for marital status, level of education and annual income.

To reduce the number of comparisons between groups in hormonal variables, we decided to calculate the magnitude of response for each hormone as well as their ratios based on the formulas derived from the trapezoidal rule [56]. Specifically, it was estimated by calculating the area under the curve with respect to the ground (AUCg) and the area under the curve with respect to the increase (AUCi). In addition, ratios were calculated using the AUC levels of each hormone.

Multivariate analysis of variance (MANOVA) was performed for hormones AUCg and AUCi, with “group” as an inter-subject factor.

Finally, linear regression models were constructed to investigate whether the IRI subscales predicted the sOXT, sT and sC responses to the task, as well as the hormonal ratios (measured by AUCi and AUCg). We ran a hierarchical regression analysis with a hormonal response as the dependent variable and IRI subscales as predictors: scores on IRI scale in Step 1; group (dummy coded as 0 for IPV perpetrators and 1 for controls) in Step 2; and the two-way interactions (IRI subscales x group) in Step 3.

Data analyses were conducted using SPSS Statistics for Windows, version 26.0 (IBM Corp., Armonk, NY, USA). Values of *p* < 0.05 were considered statistically significant. The average values are reported in the tables as mean ± standard deviation (SD).

## 3. Results

### 3.1. Participant Characteristics and Empathy Scores

There were no significant differences between IPV perpetrators and the control group in age, BMI and/or demographic variables. With respect to the IRI subscales, IPV perpetrators differed from controls in perspective taking (t (22) = −2.430, *p* = 0.024, d = 0.99) and in the fantasy subscales (t (22) = −2.635, *p* = 0.015, d = 1.07), with IPV perpetrators exhibiting lower scores than controls in both cases (see Table 1).

### 3.2. Hormonal Response to the Empathic Laboratory Task

#### 3.2.1. sOXT, sT and sC

As shown in Table 2, the MANOVA performed for independent hormone levels reported that IPV perpetrators differed in their sOXT AUCi levels compared to controls (F (1, 23) = 4.848, *p* = 0.038; η^2^_p_ = 0.181) revealing a lower sOXT AUCi value of IPV perpetrators compared to the control group. Furthermore, IPV perpetrators also differed in their sT AUCg (F (1, 23) = 4.726, *p* = 0.041; η^2^_p_ = 0.177), showing higher sT AUCg value than the control group. No further differences were found between groups in the rest of the hormonal levels.

#### 3.2.2. sOXT/T, sOXT/C and sT/C Ratios

With respect to the hormonal ratios, IPV perpetrators differed in sOXT/T AUCi (F (1, 23) = 4.385, *p* = 0.048; η^2^_p_ = 0.166) and sOXT/T AUCg values (F (1, 23) = 4.544, *p* = 0.044; η^2^_p_ = 0.171) compared with controls, with IPV perpetrators showing a lower sOXT/T AUCi and sOXT/T AUCg levels compared to the control group. Additionally, IPV perpetrators exhibited higher sT/C AUCg levels than the control group (F (1, 23) = 4.651, *p* = 0.042; η^2^_p_ = 0.175). Both groups did not differ in any other hormonal ratios (see Table 2).

### 3.3. Predictive Effects of the IRI Subscales on Hormonal Levels after the Empathic Induction Task

#### 3.3.1. sOXT, sT and sC

Only the perspective-taking subscale predicted 19.2% of the sOXT AUCi after the empathic induction task (β = 0.476, F (1, 23) = 6.45, *p* = 0.019) and 17.7% of the sOXT AUCg after the task (β = 0.461, F (1, 23) = 5.95, *p* = 0.023). Additionally, the linear regression model showed that the perspective taking predicted the 19.2% of the sT AUCi to the empathic induction task (β = −0.475, F (1, 23) = 6.401, *p* = 0.019). None of the other IRI subscales significantly predicted hormonal AUC. Moreover, no group effect was observed in this regression analysis.

#### 3.3.2. sOXT/T, sOXT/C and sT/C Ratios

Regarding the hormonal ratio levels, the linear regression model indicated that only the IRI perspective taking score significantly predicted 19.3% of the sOXT/T AUCi (β = 0.477, F (1, 23) = 6.49, *p* = 0.018). Additionally, it was seen that the IRI perspective taking score similarly predicted 20.3% of the sOXT/C AUCi (β = 0.488, F (1, 23) = 6.87, *p* = 0.016). Furthermore, perspective taking predicted 31.8% of the sT/C AUCg (β = −0.590, F (1, 23) = 11.74, *p* = 0.002). No other significant models or group effects were found.

## 4. Discussion

Our data revealed that IPV perpetrators exhibited lower scores in IRI perspective taking and fantasy subscales than controls. Furthermore, our data revealed that IPV perpetrators presented lower sOXT changes in their levels and higher total sT levels than the controls. They also presented lower sOXT/T change and total sOXT/T levels post-task, as well as higher total sT/C levels than the controls. Remarkably, the lower the perspective taking score, the lower the total sOXT levels and sOXT changes and the higher the sT increase were for the entire sample. Low perspective taking also predicted smaller sOXT/T and sOXT/C changes in the empathic induction task, and higher total sT/C levels for all participants.

Regarding the first aim, IPV perpetrators exhibited lower scores on specific IRI subscales, particularly, perspective taking and fantasy. This is in line with previous studies indicating that IPV perpetrators presented deficits in cognitive empathy functions [20,21] and linking them to IPV occurrence and severity [16]. Difficulties with perspective taking could hinder the recognition of the emotional experiences of others, increasing the likelihood of inappropriate responses in interpersonal situations [17]. It has been argued that perspective taking skills enable the consideration of various meanings and motivations, which facilitates emotional regulation. Indeed, enhanced perspective taking has been proposed as a protective factor for IPV perpetration [57]. However, no differences were found between groups in empathic concern or personal distress, even though low empathic concern and high personal distress have been linked to perpetration of IPV through an inability to tolerate the negative emotions of others [17,20,58]. Our results could be due to the susceptibility to bias (e.g., IPV perpetrators’ low emotional self-perception) and manipulation (e.g., IPV’s social desirability) of using a single self-report [6,59,60]. This outcome reinforces the importance of using empathic induction tasks as a complement to self-report questionnaires.

Secondly, we also aimed to compare the hormonal response (e.g., sOXT, sT, and sC levels and hormonal ratios) of IPV perpetrators and a control group following an empathy induction task. Lower changes in sOXT levels were observed in IPV perpetrators following an empathic induction task compared to controls. Past research pointed out that emotional videos may induce increases of OXT levels in non-violent healthy young adults of both genders [24,25]. However, the differential functioning of OXT has been proposed in IPV perpetrators. For instance, a decrease in sOXT levels was reported in IPV perpetrators after an empathic induction task involving strangers, compared to the control group that increased their sOXT levels [26]. Another study found that intranasal administration of OXT increased IPV inclinations toward partners in violence-prone individuals [61]. Indeed, despite observing no change in sOXT levels in response to an acute cognitive stress task in either the IPV or the control group, Romero-Martinez et al. [30] found that total sOXT levels were higher in IPV perpetrators. These results suggest that we cannot conclude that IPV perpetrators can be characterized by a generalized, diminished OXT functioning. In this sense, the importance of the task’s emotional content and who it involves should be highlighted. Thus, further research is needed to clarify the role of OXT in IPV proneness.

In terms of sT, both groups did not differ in their levels of sT change after the task. This was in line with a previous study, which also failed to report sT changes in IPV perpetrators to an acute laboratory cognitive stressor [30]. However, these results contrast with the decrease in sT levels found by Procyshyn et al. [25] after an empathic induction task. Additionally, Romero-Martínez et al. [20] implemented a laboratory social stressor focused on IPV and found that IPV perpetrators had higher T levels post-task than controls. Nonetheless, although no difference in change in sT levels was observed, our results exhibited higher total sT levels in IPV perpetrators relative to the control group. This finding was relevant since various authors have negatively related the performance in empathic laboratory tasks and circulating T levels. Nitschke and Bartz [62] reported that higher baseline sT levels were associated with worse performance on a naturalistic empathic accuracy task, in which participants dynamically track, in real time, the emotional state of others. In another study, Ronay and Carney [32] found that basal pT levels were negatively associated with a behavioral measure of cognitive empathy. Accordingly, it might be theorized that elevated sT levels could affect socioaffective decoding, especially in IPV perpetrators. However, we underlined the importance of considering the content and type of task when interpreting the T levels in IPV [33,63,64].

Contrary to our expectations, no significant differences in sC were detected, even though both the change in sC levels and its overall levels were lower in IPV perpetrators. This could be due to the fact that the empathic induction task is not designed to induce stress. However, it should be noted that IPV perpetrators have been seen to display lower changes in sC levels in response to a stress-inducing task compared to controls [20,30]. Hence, it is possible that the empathic induction task is related, to a greater extent, to the sOXT and sT response, not promoting a sufficient sC response to show differences between the groups. However, it is important to consider that many authors have hypothesized that individuals characterized by proneness to violence presented a hypoactivity of the hypothalamic–pituitary–adrenal (HPA) axis [65,66].

An important novelty of this study was the assessment of hormonal ratios and whether these ratios vary in response to an empathic induction task. Congruent with the above-reported results for hormone-independent levels, IPV perpetrators showed a lower change in their sOXT/T concentrations after the task, along with lower total sOXT/T levels. While the literature has provided evidence for opposing effects between these hormones in the social domain, highlighting a possible interaction between them [67,68,69], to the best of our knowledge, this is the first study to evaluate the OXT/T ratio to an empathic induction task. Notably, our findings are consistent with and strengthen the steroid/peptide theory of the social bonds (S/P Theory), which suggests that a high level of OXT (facilitator of salience of socioaffective stimuli), combined with a low level of T (facilitator of salience of self-focused stimuli), would favor socioaffective functions, such as social bonding, perspective taking, or empathy [41,70,71]. Therefore, the difference in sOXT/T ratio levels between IPV perpetrators and the control group may indicate discrepancies in the way that they process socioaffective stimuli, which could lead to an alternative empathic experience, if any.

IPV perpetrators also showed higher total sT/C levels after the task compared to the control group. Greater T/C levels have previously predicted greater aggression towards one’s partner in a non-offending sample [39]. Studies in IPV perpetrators have found higher levels of the sT/C ratio in response to social stress in this population compared to non-offender participants [20], and link sT/C to antisocial traits [33]. In fact, a high T/C ratio has been proposed to predispose an individual to self-oriented dominance attitudes [72]. Taken together, these findings reinforce our hypothesis of a different hormonal profile in IPV perpetrators when faced with an empathic induction task, with lower sOXT/T levels, related to poor socioaffective processing, and higher sT/C levels, related to a greater dominance.

Remarkably, both groups did not differ in their sOXT/C levels. A meta-analysis exposed that OXT levels may dampen C release only during challenging stress tasks that could heavily stimulate the HPA axis [35]. This lack of significance may be due to the fact that our task is not designed to induce severe social stress. However, it should be noted that the change in sOXT/C levels and the total sOXT/C levels were lower in IPV perpetrators. Future research may value exploring the OXT/C ratio in relation to empathic distress in IPV perpetrators.

Finally, we intended to explore whether empathic abilities explained the hormone levels following the empathic induction task. As a result, the perspective-taking dimension of the IRI predicted a higher sOXT response to the task, as well as a lower sT response. Consistently, greater perspective taking was also predictive of a higher sOXT/T and sOXT/C changes and a lower change in sT/C levels in response to the empathic induction task. These findings are of particular relevance as they link understanding of another person’s emotional state to an increase in sOXT after an empathic induction task. In contrast, this dimension of empathy appears, again, to be related to lower sT levels. It seems coherent that a better perception of the emotions of others was strongly related to an increase in the sOXT/T ratio, presumably through the counteracting influence of these hormones on the salience of socioaffective stimuli [27,70]. On the other hand, the predictive capacity of perspective taking with respect to the change in the sOXT/C levels could be partially explained through an attenuating effect of OXT on C when faced with the negative experiences of others [35,37], which may facilitate cognitive empathy. As a matter of fact, it has previously been observed that increased OXT predicts better cognitive accuracy following an emotional induction task, buffering C-response effects [73].

However, no other IRI dimension has shown a predictive capacity for hormone levels. This could be due to several reasons. First, difficulties have been recognized when linking the fantasy scale to empathy, being a controversial scale for several authors [74,75]. Additionally, personal distress has been primarily linked with C [76], and our task caused no change in sC levels. Finally, although OXT has been proposed to be related in part to affective empathy (closely related to empathic concern), more emphasis has been placed on the ability to recognize and understand the emotions and motivations of others (which is included in the perspective-taking dimension) [77,78]. It should be noted that no group effect was observed in the regression analyses. However, while it is true that these relationships between psychobiosocial markers were cross sectional throughout the sample, the perpetrators of IPV showed poorer perspective taking. Thus, our findings suggest that the ability to understand the situations of others could partially explain the concentration of certain hormones that are important for socioaffective processing (e.g., OXT, T and their ratio) following an empathic induction task, even in a violent population with lower empathic abilities, such as IPV perpetrators.

This study is not exempt from limitations, which are outlined as follows. First, a limitation of our study is the small sample size. This limitation is mainly due to the impossibility of collecting biological samples after the appearance of COVID-19. The COVID-19 pandemic has had and will have further consequences on biological research, and it is essential to be reminded of good practice for the management of biological samples for research, notably concerning the biosafety and security procedures [79]. This is compounded by the difficulty of access to the sample [80,81], which could be especially accentuated due to the psychobiological nature of this study. However, the study on hormones in this population is scarce, and having several hormonal determinations increases the value of the results obtained; however, further research is imperative to replicate these results in larger samples. A second limitation of our study might be the methodology used to induce empathy and elicit hormonal changes. This is a standardized task to induce negative valence emotions through the visualization of film scenes [43]. Therefore, this task is not specifically designed for empathic induction, although several studies have employed this task to induce empathy and examine OXT and T responses [24,25]. Another relevant consideration is the interpretation of the ratios. From a methodological point of view, ratio distributions are complex [82]. However, following authors’ recommendations, we considered several strategies to strengthen the statistical approach of the ratios. Nonetheless, the interpretation of the ratio findings should be viewed with caution.

Future lines of research should replicate the results of this investigation in a larger sample. In addition, it would be of great interest to explore possible differences between IPV perpetrators and other types of offenders, helping to establish risk factors that may be transversal to violent behavior and specific risk factors according to the crime committed. Additionally, further research covering not only the response to negative effects, but also to positive effects, seems necessary. This would allow for a better understanding of the processing of positive emotions in IPV perpetrators. In this regard, it would be of great interest to address emotions associated with prosocial behavior, an important protective factor for violent behavior, such as compassion. Future research could also be conducted to analyze the empathic response of IPV perpetrators when focusing on aggressors rather than victims, and whether their empathic response could be related to the offender accountability. Finally, exploring the effects of IPV intervention programs on psychological and hormonal response following an empathic induction task could highlight the importance of adequate emotional processing for IPV prevention.

## 5. Conclusions

Our results seem to indicate that IPV perpetrators exhibit both biological and psychological markers related to empathy difficulties. Specifically, they show lower levels of sOXT over sT in response to an empathic induction task that predisposes subjects, presumably, to differential processing of socio-affective stimuli. This disparity may be based on self-oriented dominance attitudes. The interaction between OXT and T appears to be relevant for a holistic understanding of the empathic response, given that important socio-affective functions for empathy, such as perspective taking, have been related to a higher OXT/T ratio. Thus, future lines of research in this direction seem pertinent. All of this, together with lower self-reported empathy scores, may point to an abnormal empathic function that would act as a risk factor for IPV. Taking this into account, our results suggest that addressing certain dimensions of empathy in intervention programs could influence hormonal markers related to reduced dominance attitudes, better anxiety management, and increased attention to social-affective stimuli, which could ultimately facilitate the modification of aggressive behavior patterns [83].

## Figures and Tables

**Table 1 ijerph-19-07897-t001:** Means, standard deviations, percentages, and means comparisons for socio-demographic and psychological variables for all groups.

	IPV (*n* = 12)	Controls (*n* = 12)	Chi-Square	*p*-Significance
Marital status (%)				
Married	58	50	0.17	0.682
Single/Divorced/Widowed	42	50		
Level of education (%)				
Primary	08	08	3.14	0.208
Upper secondary	75	42		
University	17	50		
Annual Income (%)				
Low income	17	25	8.01	0.333
Medium income	67	67		
High income	16	08		
Age *(M, SD)* BMI *(M, SD)*	**IPV (*n* = 12)**	**Controls (*n* = 12)**	** *t* ** **-test independent samples**	**Cohen’s d**
	35.17 (8.11)	40.25 (13.87)	−1.09	0.45
	25.83 (3.66)	24.38 (3.33)	1.01	0.41
IRI Perspective Taking *(M, SD)*	21.08 (3.18)	25.00 (4.59)	−2.43 *	0.99
IRI Empathic Concern *(M, SD)*	25.67 (4.52)	27.33 (5.66)	0.80	0.32
IRI Personal Distress *(M, SD)*	15.83 (4.39)	15.75 (4.45)	0.05	0.02
IRI Fantasy *(M, SD)*	18.42 (5.05)	23.33 (4.03)	−2.64 *	1.07

Note: IPV—intimate partner violence; BMI—body mass index; IRI—interpersonal reactivity index; M—mean; SD—standard deviation. Statistical significance * *p* < 0.05.

**Table 2 ijerph-19-07897-t002:** Means, standard deviations, percentages, and MANOVA for the hormonal AUCg and AUCi for all groups.

	IPV (*n* = 12)	Controls (*n* = 12)	F Value	Partial Eta Square
sOXT AUCg *(M, SD)*	538.13 (41.40)	552.76 (40.91)	0.76	0.033
sT AUCg *(M, SD)*	415.40 (34.79)	384.87 (34.03)	4.73 *	0.177
sC AUCg *(M, SD)*	139.25 (37.72)	146.84 (64.81)	0.12	0.006
sOXT AUCi *(M, SD)*	−25.92 (79.30)	43.02 (73.99)	4.85 *	0.181
sT AUCi *(M, SD)*	−0.43 (37.97)	−6.97 (27.29)	0.24	0.011
sC AUCi *(M, SD)*	−6.59 (25.15)	6.92 (49.52)	0.71	0.031
sOXT/T AUCg *(M, SD)*	52.33 (60.29)	99.58 (47.56)	4.54 *	0.171
sOXT/C AUCg *(M, SD)*	−361.92 (45.17)	−358.00 (72.94)	0.03	0.001
sT/C AUCg *(M, SD)*	−414.75 (36.90)	−452.67 (48.45)	4.65 *	0.175
sOXT/T AUCi *(M, SD)*	−19.38 (85.58)	47.18 (69.28)	4.39 *	0.166
sOXT/C AUCi *(M, SD)*	−12.81 (67.06)	19.18 (82.05)	1.09	0.047
sT/C AUCi *(M, SD)*	6.19 (40.26)	13.95 (35.98)	1.67	0.071

Note: sOXT—salivary oxytocin; sT—salivary testosterone; sC—salivary cortisol; M—mean; SD—standard deviation; MANOVA—multivariate analysis of variance. Statistical significance * *p* < 0.05.

## Data Availability

The data presented in this study are available on request from the corresponding author. The data are not publicly available due to the presence of sensitive information from convicted persons.

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
