# Peer review of "Hormonal Profile in Response to an Empathic Induction Task in Perpetrators of Intimate Partner Violence: Oxytocin/Testosterone Ratio and Social Cognition"

_ijerph, 2022, doi:10.3390/ijerph19137897_

Round 1

Reviewer 1 Report

Thank you for the opportunity to review the manuscript titled "Hormonal profile in response to an empathic induction task in perpetrators of intimate partner violence: Oxytocin/Testosterone ratio and social cognition"

This manuscript examines two groups of people, 12 individuals who are identified as perpetrators of  intimate partner violence, and 12 control individuals.  Several variables are identified as being associated with proneness to perpetrate intimate partner violence.

The background is complete and well presented, as are the methodology and results sections.  Conclusions follow logically from the experiment results.

However, a major shortcoming with this study is that it fails to do an a priori power analysis.  given that only 24 total participants are examined, the odds of an erroneous result exponentially increase with each statistic.  Tables 1 and 2 include 16 independent t-tests related to outcome variables, which do not include the t-tests and chi-square tests for comparing relevant data comparing the comparability between the experimental and control groups.  Finally, linear regression models  were also included to examine the data collected.

Given the number of analyses, it is not possible to tease out if the significant results were erroneous and based on a Type I error.  My recommendation is to rerun the data collection by performing appropriate power analyses, include a more sizeable N, and reduce the number of pairwise comparisons (t-tests).

Reviewer 2 Report

This is a challenging paper, with some interesting results.  This reviewer is not an expert in this field, so he cannot comment on  the methodology of measuring aspects of physiology, which must be taken on trust.

The reviewer is surprised that such strong significance was obtained with such a small number of subjects. This means that the study should be replicated as soon as possible, since this may have been an idiosyncratic sample. The authors should make it clear that replication work is needed.

Author Response

Reviewer 2

Reviewer #2: This is a challenging paper, with some interesting results.  This reviewer is not an expert in this field, so he cannot comment on the methodology of measuring aspects of physiology, which must be taken on trust.

Response. We are very grateful for the reviewer’s positive comments. We thank him/her for his/her effort and valuation of our paper, which is very encouraging and motivates us to continue our research in this field.

The reviewer is surprised that such strong significance was obtained with such a small number of subjects. This means that the study should be replicated as soon as possible, since this may have been an idiosyncratic sample. The authors should make it clear that replication work is needed.

Response. As suggested, we have reinforced the section on limitations in relation to sample size. In this regard, we have emphasized the constraints and obligations of biological research associated with the COVID-19 pandemic. Furthermore, we have integrated the added difficulty of accessing intimate partner violence perpetrators, whose intervention programs were suspended during the COVID-19 pandemic. Finally, we have stressed that it is imperative to replicate these results in larger samples to provide greater validity to our study. (Lines 446-455)

Reviewer 3 Report

Congratulations to the authors for this well designed study.

The paper is detailed, clear, well organized and written. The method is clear. The authors undelined the limitations of the study and suggestions for future lines of research.

I report below some little suggestions.

Introduction:  In the Introduction section the authors could provide a definition (or description) of what is an induction task.

The paper clearly provides an overview of previous literature. It also clearly describes the study aims.

Methods and Results: congratulations. Methods and measures are very clearly described, as well as the Results section.

Discussion: line 317-319: Could the results also be due to the low numerosity of samples?

Line 335-337: A further aspect to investigate could be who the participants identify with during the task (perpetrator, victim, both). May be useful to ask this aspect after the task. What do the author think about this?

Some grammar errors should be revised throughout the text, for instance line 52.

Author Response

Reviewer 3

Reviewer #3: Congratulations to the authors for this well-designed study. The paper is detailed, clear, well organized and written. The method is clear. The authors underlined the limitations of the study and suggestions for future lines of research. I report below some little suggestions.

Response. Our research team appreciates the evaluation received from the reviewer. We thank him/her for his/her feedback and his/her assessment of our work, which is very positive and motivates us to continue our research in this field.

Introduction:  In the Introduction section the authors could provide a definition (or description) of what is an induction task.

Response. As suggested by the reviewer, we have added a description of an empathic induction task to increase the readability and clarity of our empathic induction task (described in depth in the "methodology-procedure" section). We hope that this new version is adequate. (Lines 52-60)

The paper clearly provides an overview of previous literature. It also clearly describes the study aims. Methods and Results: congratulations. Methods and measures are very clearly described, as well as the Results section.

Response. We highly value this comment, as we have placed special attention in clearly and correctly describing the methodology and the results obtained from it.

Discussion: line 317-319: Could the results also be due to the low numerosity of samples?

Response. As mentioned by the reviewer, the sample size is a limitation to be taken into account when interpreting the results. For this reason, we have emphasized the importance of replicating our results in a larger sample. In addition, we emphasize that difficulties associated with the COVID-19 pandemic continue to have a direct impact on the current situation in biological research. This is compounded by the difficulty of access to the sample, which may be especially accentuated due to the psychobiological nature of this study. (Lines 446-455)

Line 335-337: A further aspect to investigate could be who the participants identify with during the task (perpetrator, victim, both). May be useful to ask this aspect after the task. What do the author think about this?

Response. Following this observation, we have integrated in the "future research" paragraph the relevance of analyzing the empathic response of intimate partner violence perpetrators when they are focused on the aggressors and not on the victims, and whether their empathic response could be related to the offender accountability. (Lines 472-475)

Some grammar errors should be revised throughout the text, for instance line 52.

Response. Concerning this commentary, we have carefully revised the grammatical errors throughout the text. We trust that this revised version will be adequate.

Round 2

Reviewer 1 Report

Thank you for the opportunity to re-review this manuscript, and I am thankful to the authors for taking the time to review my recommendations and concerns about the manuscript.  It is clear that a great deal of time and effort has gone into preparing this manuscript for publication, and I am impressed in the attention to detail the authors have put into describing their methodology and data analysis.

My initial concerns about this manuscript had to do with the lack of power given a small sample size and repeated statistical analyses.  The authors correctly observe that utilization of the MANOVA statistic provides multiple simultaneous analyses with a single statistic, which is appropriate for this research design.  However, the manuscript continues to include multiple separate t-test analyses in addition to the MANOVA, which increases the likelihood of erroneous significant results.  Further, I did not find the term "MANOVA" mentioned at all in the revised manuscript, and explicitly identifying the statistic may make it easier for consumers to understand that the authors were concerned about power during their analyses.  

The authors discuss the difficulty with accessing the sample in this research, and I am extremely sympathetic to this concern.  However, this does not alleviate the concern about power and appropriate analyses.  This concern was raised by at least one of the other reviewers as well.  My suggestion would be to consider leaving the t-test analyses out of the manuscript entirely, and focus solely on the MANOVA analysis, while maintaining the discussion of limitations due to small sample size.  As the other reviewers have remarked, the manuscript, as-written, raises serious concerns about replicability due to small sample size and multiple statistics employed.
